# **GCInsights: Consistency in Pyrocartography Starts With Colour**

# Benjamin J. Hatchett<sup>1,2</sup>

<sup>1</sup>Cooperative Institute for Research in the Atmosphere, Colorado State University, Ft. Collins, Colorado, USA <sup>2</sup>National Oceanic and Atmospheric Administration, Global Systems Laboratory, Boulder, Colorado, USA **Correspondence:** Benjamin Hatchett (Benjamin.Hatchett@gmail.com)

**Abstract.** Fire progression maps provide operational and public information regarding wildland fire spread, size, and proximity to critical assets through time. Cartographic guidance regarding the use of colour to denote the sequential nature of fire progression is limited, leading to inconsistency in fire progression maps produced for operational, research, and public applications, which potentially limit these map's accessibility and ability to effectively communicate information. In this paper, I provide colourmap recommendations to facilitate consistent, intuitive, and accessible fire progression mapping.

## 1 Introduction

The movement of wildland fire across the landscape results from factors including weather, fuels, topography, fire history, and fire suppression activities. The most extreme cases of fire spread occur when strong winds and the orientation of topography align with accumulations of continuous fuels available to burn at high intensity and produce downstream ignitions–spot fires–

10 meters to kilometers ahead of the flaming front. Fire progression maps (FPMs) provide a first-order visualization of a wildfire's current perimeter and spread since ignition (Figure 1). Typically, FPMs are produced daily using aircraft-based infrared data, though near-real estimates are increasingly available using satellite data (Chen et al., 2022; Berman et al., 2023; Liu et al., 2024).

Operationally, FPMs assist fire managers in formulating strategies and implementing tactics for achieving desired manage-15 ment outcomes and to understand prior management efforts. Forecast FPMs produced by fire behavior models using existing fire perimeters or ignition locations support fire management by simulating fire spread possibilities and potential smoke production (Kochanski et al., 2023). Public information officers and agency websites (e.g., https://inciweb.wildfire.gov/) share FPMs with the public, media, and agency partners to provide updates on fire activity and suppression efforts. Researchers and Burned Area Emergency Response teams may use FPMs to evaluate when a particular area burned, since the time of burning in

a particular location often does not coincide with the ignition or containment date. Simulated FPMs extend to pre-fire planning efforts to help prioritize fuel management strategies and locations by mapping potential growth given varying treatments and ignition locations. Taken together, the creation and dissemination of FPMs for applications throughout the fire cycle motivates the need for maps that communicate effectively.

### 2 Current Guidelines, Current Challenges

- To provide a basis for consistency in geospatial products during wildfire incidents, the U.S.-based National Wildfire Coordinating Group (NWCG) maintains a set of cartographic standards called the NWCG Standards for Geospatial Operations "GeoOps" (Publication Management System 936) (National Wildfire Coordinating Group, 2024). GeoOps provides guidance on map product standards for wildland fire mapping with two specific recommendations for FPMs. First, it recommends using "standardised" colour ramps (or colourmaps) to show trends instead of discrete values when showing more than five time steps.
- Second, it recommends the standard element of the fire perimeter data for each time period, transitioning from cool (older) to warm colours (more recent).

Despite this guidance, at the time of writing, the GeoOps examples demonstrate known challenges in visual communication: the use of inconsistent (i.e., "standardised" is not defined explicitly) colournaps that are potentially inaccessible for colour vision deficient users to portray fire progressions (Figure A1). This highlights a missing, but easily remedied, aspect in GeoOps:

- the recommendation for sets of colourmaps that improve accessibility (Crameri et al., 2020; Stoelzle and Stein, 2021) and address 508 Compliance, a U.S. Federal law enacted to create and maintain standards enabling the accessibility of electronic and information technology (i.e., web-based content or multimedia such as portable document formats) to those with disabilities (United States Congress, 1973). Further, the lack of colourmap consistency in FPMs (cf. Figure A1 and those in Chen et al., 2022, Kochanski et al. 2023, and Liu et al. 2024) represents another potential limiting factor in user cognition for maps already
- displaying complex information (Bunch and Lloyd, 2006) if the colourmaps change substantially from day-to-day, incidentto-incident, or across applications.

#### **3** Colourmap Recommendations

Near-daily fire perimeters using airborne infrared data from California's 2021 Dixie Fire (13 July–27 September 2021; 389,824 ha (contained on 25 October 2021)) and 2013 Rim Fire (17 August–24 October 2013; 104,131 ha) serve to demonstrate
alternative colourmaps that address aforementioned limitations (Figures 1 and B1). Perimeter data was acquired from the National Interagency Fire Center (http://www.ftp.wildfire.gov). For clarity, I omit other standard cartographic elements of FPMs (key geographic features including topography, hydrography, and roads).

Fire progression across space over time is an ordinal process, implying sequential maps (Figure 1a-b) are more logical than diverging maps (Figure 1c). Further, mapping temporal data enhances the complexity of the display, implying the use

- of diverging colourmaps to show sequential processes may cause users to misinterpret or miss important changes (Buckley, 2017; Crameri et al., 2024). As the GeoOps notes, fire progressions include a thermal component as recently burned areas demonstrate intuitively higher temperatures. This motivates the cool-to-warm sequence, and whether such colourmaps should preserve intuitive associations (i.e., cool blue-to-hot red) instead of physically-consistent associations (i.e., black-body emission temperatures) remains open to discussion. Here, I select the sequential colourmap "YIOrRd" (yellow-orange-red; reversed to
- mimic the blackbody radiation colour curve) from Colorbrewer (Brewer et al., 2003) and the colourmaps "Batlow" and reversed "Managua" (Crameri, 2021) as suggested alternatives. While the sequential colourmaps "YlOrRd" and "Batlow" print well in