# Peer review of "GCInsights: Consistency in Pyrocartography Starts With Colour"

_Geoscience Communication, 2024_

## Referee Comment (RC1)

a) Turbo

b) Turbo with isochron lines

Fire
Origin

Fire
Origin

Time ➡

Time ➡

c) Batlow

d) Batlow with isochron lines

Time ➡

Time ➡

e) Reversed YlOrRd

f) Reversed Managua with isochron lines

Time ➡

Time ➡

a) Turbo

b) Turbo with isochron lines

Fire Origin

Fire Origin

Time →

Time →

c) Batlow

d) Batlow with isochron lines

Time →

Time →

e) Reversed YlOrRd

f) Reversed Managua with isochron lines

Time →

Time →

a) Turbo

b) Turbo with isochron lines

Fire
Origin

Time ➝

Fire
Origin

Time ➝

c) Batlow

d) Batlow with isochron lines

Time ➝

Time ➝

e) Reversed YlOrRd

f) Reversed Managua with isochron lines

Time ➝

Time ➝

a) Turbo

b) Turbo with isochron lines

Fire
Origin

Fire
Origin

Time ➝

Time ➝

c) Batlow

d) Batlow with isochron lines

Time ➝

Time ➝

e) Reversed YlOrRd

f) Reversed Managua with isochron lines

Time ➝

Time ➝

---

## Author Comment (AC1)

**To:** Reviewer No. 1
**Re:** Responses to Reviewer 1 (Dr. Fabio Crameri) Comments
**Date:** 19-Feb-2025
* * *
Dear Dr. Crameri,

Thank you for your constructive and beneficial but also thought-provokingly critical suggestions for revisions. Your suggestion (and explanation of why it was a major issue) to omit the "Turbo" colormap as well as the suggested idea (shared also by Reviewer 2) to alter Figure 1 and show colourblindness simulations of suggested colourmaps instead of multiple examples of the same colourmaps were particularly appreciated. I have revised the manuscript to follow your suggestions and edits, leading to improvements in content and clarity throughout. My responses are in red text with newly added or revised text in *italics* using **bold for emphasis**. I greatly appreciate your time and energy in reviewing our manuscript and providing your thoughtful perspective.
Sincerely,
Ben Hatchett
* * *
Reviewer 1 Summary: Inconsistencies and misusage of colour in pyrocartography (and beyond) is a major issue and needs to be tackled. This manuscript nicely tackles this problems and provides a clear solution to make understanding, communicating, and tackling fire hazard more accurate, effective, and accessible. I therefore think that it is a very valuable contribution to be published in Geoscience Communication. The article is clearly presented and only has some minor misrepresentation that I think should be fixed before publication.

Thank you for your positive remarks and for your highly constructive, thoughtful, and thorough review of the manuscript. All of this input improved the manuscript and I learned several new things.

I have one major issue: The colour map "Turbo" is presented and used as a perceptually uniform (and I guess colour blind friendly) colour map, which it is not. I think it should be omitted as recommendation here (and elsewhere) to avoid further misconception about its properties. See a more detailed explanation below.

I appreciate the issue being raised about my initial misrepresentation of the 'Turbo' colormap. Following the reasoning provided by your comment, I have **removed** the 'Turbo' colormap.

I have one major suggestion: Would simulating and providing colour-vision deficient appearance (as is done for figure A1), and maybe even greyscale conversion, of figure 1 be useful for the purpose of the manuscript? – I think it would and attached some simulations (based on Brettel et al., 1997; happy to share adjusted figures for the author). That way, it would also become more clear that Turbo is not suited for a scientific application (repeated colours along the scale).

This is an excellent suggestion, which was also provided by reviewer 2, thus highlighting the value of performing the colourblindness simulation. I have followed this suggestion as it will

more strongly help to make the case of the manuscript as both reviewers noted. To keep the figure to a reasonable size, I opted to include the three colourmaps with no isochron (top row; given this is a noted suggestion in the main text) and with isochrons (next row) and show two colour-vision deficient appearances in the following two rows (deuteranopia and protanopia, respectively). Following the suggestion of reviewer 2, I include only one burn area (the Dixie Fire) in the main manuscript and moved the second example (the Rim Fire) to the appendix. I left out the greyscale simulation since the figures were becoming small but added a note to the text that the sequential maps work well for greyscale but the diverging Managua requires a fire origin and/or labeled isochrons to orient the direction of fire progression. New text:

*"While the sequential colourmaps "YlOrRd" and "Batlow" print well in black and white, the diverging "Managua" requires additional annotation (e.g., an origin point or labeled isochrons) to orient readers to the direction of fire progression."*

The new figures and their revised captions are shown on the next page:

[Figure]

Figure 1: Revised Figure 1 (a-c) Daily fire progression maps of the 2021 Dixie Fire using three colourmaps that are accessible for colour vision deficient viewers and demonstrate a physically-intuitive sequential progression through time (i.e., older shown by cooler cooler colours and newer by warmer colours. (d-f) As in (a-c) but including isochrons. (g-i) Maps in (d-f) with deuteranopia (green-blind) colourblindness simulation. (j-l) Maps in (d-f) with protanonopia (red-blind) colourblindness simulation. The yellow star denotes the fire origin location. *Note the missing 'e)' label, I will fix this in the revised manuscript.

[Figure]

Figure 2: New Figure B1: (a-c) Daily fire progression maps of the 2013 Rim Fire using three colourmaps that are accessible for colour vision deficient viewers and demonstrate a physically-intuitive sequential progression through time (i.e., older shown by cooler cooler colours and newer by warmer colours. (d-f) As in (a-c) but including isochrons. (g-i) Maps in (d-f) with deuteranopia (green-blind) colourblindness simulation. (j-l) Maps in (d-f) with protanonopia (red-blind) colourblindness simulation. The yellow star denotes the fire origin location.

Reviewer 1 Specific Comments:

Line 32: Please clarify what "non-colorsafe colormaps" means.

I apologize for the imprecise language here. I have modified the text to be consistent with the medical terminology for colorblindness (National Eye Institute of the National Institutes of Health, see: `https://www.nei.nih.gov/learn-about-eye-health/eye-conditions-and-diseases/color-blindness` to now state these (currently recommended) maps "are potentially inaccessible for colour vision deficient users".

*...demonstrate known challenges in visual communication: the use of inconsistent (i.e., "standardised" is not defined explicitly) colourmaps **that are potentially inaccessible for colour vision deficient users** to portray fire progressions...*

Lines 45–47: To support these statements, we clarify the use of different colour gradient types in: Crameri, F., G.E. Shephard, and P.J. Heron (2024). Choosing suitable color palettes for accessible and accurate science figures. Current Protocols, 4, e1126. https://doi.org/10.1002/cpz1.1126

Thank you for the reference suggestion to shore up this statement. This reference also helped me point out the interesting case of "Managua", which may be an exception to the misuse of diverging maps for sequential processes (given it "diverges" logically from cool (blue) to warm (red) to hot (yellow). Before this statement, I did revise the text to note a diverging map does need a temporal reference point (fire origin) to orient readers, especially if black and white is used (i.e., in printing). New text:

"*The physically-intuitive nature of "Managua" (cool-to-warm-to-hot) also may allow it to show sequential fire progressions despite being a diverging colourmap and thus an exception to the guidance provided in Crameri et al., 2024).*"

Line 47: Put the Buckley, 2017 reference into parentheses (i.e., "\citep{xxx})"– same problem in other instances, such as line 51

Changes made, thank you for catching this.

Lines 48–50: Good point!

Thanks! :-)

Line 50: "We" should be "I" for this single-author paper, shouldn't it?

Indeed, changes have been made here and in another instance.

Line 52: Turbo is NOT perceptually uniform. It is confused widely as such though. I guess there is less scrutiny for a colour map developed by a big company than by individual scientists. Turbo is better than a standard rainbow (such as Jet) in terms of perceptual evenness, but it is still not on par with e.g., batlow or YlOrBr, and definitely not perceptually uniform (see e.g., the author's clarification that "[Turbo] is not a perceptually linear" and "[Turbo is intended] for day-to-day tasks where perceptual uniformity is not critical" on `https://www.google.com/url?sa=t&source=web&rct=j&opi=89978449&url=https://research.google/blog/turbo-an-improved-rainbow&ved=2ahUKEwjE4snyxbuKAxUjgP0HHU1jOBQQFnoECBoQAQ&usg=AOvVaw3ZCRKST1BPJ1AX72-lwrl4).`

This is visible in Figure 1, where the neighbouring colours in the blueish parts of the color bar clearly differ more strongly than e.g., in its greenish parts.

I appreciate the additional clarification and the solid points raised here. Per the major issue raised early on in the review, I have removed "Turbo" from use in the suggested colourmaps and now show "YlOrRd", "Batlow", and "Managua".

Line 54: The Ware et al., 2023 paper is trying to make a case for the use of non-perceptually-uniform and inaccessible colour maps, so it does not fit in here, I think.

Agreed, this reference has been removed here.

Lines 57–59: Not sure that is a helpful statement. It is unclear what "situation-specific adjustments to colormaps" means. If it means distorting the uniformity of color gradient of the scale, than this is a bad suggestion, as it would distort the actual data, and for the reader impossible to reproduce (or judge for its validity). In particular in plots showing the spread of fire, it seems key to properly display where the fire spread slowly, and how much more rapidly it spread elsewhere. Distorting the colour scale would suppress this information and misrepresent it.

This is a solid and thoughtful critique, which I appreciate and agree with. As such, I have removed this sentence from the revised manuscript.

Lines 61–63: Along similar lines, I disagree with this statement, as either a colour map is perceptually uniform or it is not. What "used appropriately" and "better design" mean here is unclear. Moreover, the paper referenced argues with very limited cases, and provides suggestions that are implementable only by visualisation experts and rather confusing to everyone who is not. And let's remember, hardly any scientist has received even the basic education in scientific visualisation. The paper's argument can be mentioned here, but I don't think it is fair nor clear to the reader to present it equally to other, more broadly applicable studies and more broadly based arguments.

These are extremely valid points and I appreciate your critical take on the use of my unclear language. I agree, and as such I removed the final argument and moved the previous sentence noting the use of rainbow/Turbo in fire progression mapping and scientific literature more broadly to the conclusion. To your point about lacking basic education in scientific visualization, I could not agree more and I wish this aspect was integrated more deeply into data analysis courses as well as made part of our general curriculum in scientific critique at undergraduate and graduate levels (to say the least).

Figure 1: Given that many people tend to look at figures mostly, these days, the figure caption could be a bit more descriptive to be more helpful: for example, explain that these are recommended colour maps, and what techniques are used in (b,d,f) to increase accessibility.

Agreed, I have added additional description pursuant to the reviewer's suggestion to improve the value of the caption text. Please see the revised captions associated with the revised Figure 1 and new Appendix Figure 2.

I enjoyed reading this nice piece and would like to thank the author for their effort!

I appreciate this statement, and again, thank you for your review!

And finally, for transparency, I am the author of the Scientific colour maps (some of which are shown here).

---

## Author Comment (AC2)

**To:** Reviewer No. 2
**Re:** Responses to Reviewer 2 (Dr. Richard Westaway) Comments
**Date:** 19-Feb-2025
* * *
Dear Dr. Westaway,

Thank you for your detailed review and your thoughtful and helpful suggestions for revisions. Your suggestions to re-work Figure 1 to better highlight the colourblindness simulations to better support the narrative of the paper was particularly valuable and appreciated. Your additional suggestions and detail-oriented edits led to improvements throughout. My responses are in red text with newly added or revised text in *italics* using **bold for emphasis**. I very much appreciate your taking the time to review my manuscript and providing your insights.

Sincerely,

Ben Hatchett
* * *
Summary

The author explains the utility of fire progression maps (FPM) and argues, given their importance, that greater consideration should be given to the development and adoption of standardised colormaps to portray fire progressions. The paper is well argued, clearly written and introduces an interesting application where the visual communication of spatial information is of great importance to a range of stakeholders. Since I have no prior experience in pyrocartography, the following comments are from a more general colour accessibility and science communication perspective rather than written with any insight in this particular application.

Thank you for the positive words and enthusiastic support of the work as well as your comments and corrections.

General comments

I have two general comments:

(1) Figure 1

This figure is intended to illustrate the importance of colormaps for FPMs, and also serves to demonstrate some different color options that are available. However, I find the choice of datasets and colormaps as currently presented is somewhat muddled and suggest that an alternative set of panels might more clearly support the narrative. Currently, Figure 1 presents fire perimeter data from two fire events, demonstrates four different colormaps as well as showing the visual impact of including isochrons. However Figure 1 does not currently include any color blindness simulation or black and white printing challenges (as are shown in Figure A1 for other fire progression mapping examples).

While not possible to show every permutation, I would suggest simplifying what is being shown for example by separating (or only showing one of) the two fire events – plus adding in some examples of colour-vision deficient appearance – such that 12 panels are shown for each fire event

(3 panels wide x 4 panels tall), with the panels progressing more logically and allowing easier comparison: 1a Rim Fire Turbo; 1b Rim Fire Turbo with isochrons; 1c Rim Fire Turbo with color blindness simulation; 1d Rim Fire Batlow; 1e Rim Fire Batlow with isochrons; 1f Rim Fire Batlow with color blindness simulation; 1g Rim Fire reversed YlOrBr; 1h Rim Fire reversed YlOrBr with isochrons; 1i Rim Fire reversed YlOrBr with color blindness simulation; 1j Rim Fire reversed Managua; 1k Rim Fire reversed Managua with isochrons; 1l Rim Fire reversed Managua with color blindness simulation. Then I would suggest repeating for the Dixie Fire, i.e.: 2a Dixie Fire Turbo; 2b Rim Fire Turbo with isochrons; 2c Rim Fire Turbo with color blindness simulation; and so on...

Recognising the figure constraints of the GCInsights format, and given that the full 24 panels would be near impossible to display together, I would advocate illustrating perhaps one of the fire events in the main body text (Figure 1 a-l) and moving the other to the Appendix (new Figure A2 a-l). If this is still felt to be too many panels to communicate clearly, my instinct and preference would be that color vision deficient appearance is more important to illustrate (i.e. has a larger effect on visual communication) than the presence/absence of isochrons, which would eliminate the need for four panels per fire event.

Thank you for the excellent and thoughtful ideas to adjust Figure 1, which reviewer 1 also noted as a major concern. By merging both of the reviewer suggestions, I have revised Figure 1 to now show a single fire (the Dixie Fire) and to include color vision deficient simulations (deuteranopia and protanopia, respectively) for three 'better' colormaps. Following the suggestion here, I include the second fire (the Rim Fire) as an additional appendix figure.

I wish to note that following suggestions from reviewer 1, the "Turbo" colormap has been removed and now both the revised Figure 1 and new Appendix Figure 2 show the same colormaps. At this stage, I opted to include the isochron comparison as I feel this is an important point made in the text as well (to help delineate the "just noticeable differences" that arise as discrete colormaps grow in the number of steps shown to begin to approach a continuous colormap. The two new figures are shown below; note these include isochrons but no black/white simulation; the diverging nature of the Managua option limits its ability to be assessed without labeled isochrons or without a fire origin point, this is now noted in the text. Also per the next comment, please note additional details are provided in the captions:

[Figure]

Figure 1: Revised Figure 1 of the 2021 Dixie Fire (note the same version of the Rim Fire is now shown as Figure A2. The revised caption is explained in response to the next comment.

[Figure]

Figure 2: New Figure A2 of the 2013 Rim Fire. The revised caption is explained in response to the next comment.

The figure captions should also be changed accordingly, and I would suggest expanded to explain each panel more fully.

Agreed, and a similar point was raised by reviewer 1. Here are the revised, expanded captions for the new Figure 1 and Appendix Figure 2:

Caption for revised Figure 1: "*(a-c) Daily fire progression maps of the 2021 Dixie Fire using three colourmaps that are accessible for colour vision deficient viewers and demonstrate a physically-intuitive sequential progression through time (i.e., older shown by cooler cooler colours and newer by warmer colours. (d-f) As in (a-c) but including isochrons. (g-i) Maps in (d-f) with deuteranopia (green-blind) colourblindness simulation. (j-l) Maps in (d-f) with protanonopia (red-blind) colourblindness simulation. The yellow star denotes the fire origin location.*"

Caption for new Appendix Figure 2: "*(a-c) Daily fire progression maps of the 2013 Rim Fire using three colourmaps that are accessible for colour vision deficient viewers and demonstrate a physically-intuitive sequential progression through time (i.e., older shown by cooler cooler colours and newer by warmer colours. (d-f) As in (a-c) but including isochrons. (g-i) Maps in (d-f) with deuteranopia (green-blind) colourblindness simulation. (j-l) Maps in (d-f) with protanonopia (red-blind) colourblindness simulation. The yellow star denotes the fire origin location.*"

(2) Preferred colormap(s)

Given the obvious thought that the author has given to the issue of colormaps for FPMs, and the suggestion in the abstract that the paper provides "colormap recommendations", it might be helpful to readers if the final section of the paper included some specific recommendations for which of the colormaps presented might provide the best basis for standardisation (or alternatively if any of the colormaps presented are clearly less suitable to use in this context). While Figure 1 presents various different colormap and map presentation options, the author does not currently provide any views or evaluation on their relative ability to communicate the desired information. The inclusion of color vision deficient maps in Figure 1 (as suggested above) might help demonstrate how some of the colormaps shown are perhaps less suitable than others (e.g. repeated colors in Turbo).

I appreciate the request to further weigh in on what colourmap to suggest as ideal, and to address this suggestion I have added some text to the conclusion but nonetheless must "leave the door open" for the broader community to also weigh in on. The colourmaps provided were selected based upon their meeting the fundamental necessities of visual accessibility (perceptual uniformity, colourblind-accessible) and their sequential and intuitive nature of older as cooler and more recent burning as warmer colours. This input from the community would likely be greatly facilitated by the integration of social science and usability testing with varied target audience and user groups ranging from the general public to fire management official to qualitatively and/or quantitatively. Social science-based approaches such as those applied by Morrison et al. (2024, Int. J. Wildland Fire) to maps specifically or more broadly to wildfire decision support systems (which may include a model-based fire behavior/spread/progression component such as Fire Spread Pro in the Wildland Fire Decision Support System evaluated by Nobel and Paveglio (2020; J. Forestry) demonstrates ways to perform these assessments.

New text noting both why I selected these (coloursafe, perceptually uniform, sequentially, and

following the cold-to-hot progression of fire) and an idea way to have the broader community work towards a standard by employing social science methods (in reverse order since that seemed to flow better from 'here's some things we as a community should do, but here's why I picked these as starting points'):

*"Ideally, collaborative efforts between users and producers of FPMs would integrate social science-based methods to robustly identify the needs, preferences, usability and accessibility of maps as informational and/or decision support tools across varied end-user audiences from the general public (e.g., Morrison et al., 2024) to operational fire managers (e.g., Noble and Paveglio, 2020). By providing examples meeting contemporary visualization standards (e.g., accessible for colour vision deficient users and demonstrating perceptual uniformity) and intending to be intuitive (sequential and physically-consistent), the three suggested colourmaps (Figure 1 and A2) intend to provide a starting point for such efforts."*

Specific comments

1. p1, line 4-5: I would suggest the end of the abstract is reworded to "...applications, which potentially limit these map's accessibility and ability to effectively communicate information. In this paper, I provide colormap recommendations to facilitate consistent, intuitive, and accessible fire progression mapping."

Thank you for the suggestion, I have followed your recommendation and revised the text in the abstract accordingly.

2. p1, line 6: Change "Wildland fire's movement.." to "The movement of wildland fires…"

Change made, thanks.

3. p1, line 16: Unnecessary comma after parentheses.

Good catch, comma removed.

4. p2, line 31: Define or explain "non-colorsafe" (and hence "color-safe" on line 34)

Apologies for the imprecise language. I have modified the text to be consistent with the medical terminology for colorblindness (National Eye Institute of the National Institutes of Health, see: https://www.nei.nih.gov/learn-about-eye-health/eye-conditions-and-diseases/color-blindness to now state these (currently recommended) maps "are potentially inaccessible for colour vision deficient users". New text:

*"...demonstrate known challenges in visual communication: the use of inconsistent (i.e., "standardised" is not defined explicitly) colourmaps **that are potentially inaccessible for colour vision deficient users** to portray fire progressions (Figure A1)."*

5. p2, line 31 and 34: Inconsistent spelling of colorsafe/color-safe

Revised for consistency, thank you for pointing this out.

6. p2, line 34: Define or explain "508 Compliance"

Thank you for this request, some additional detail has been added about the goals of 508 Compliance:

"*...and address 508 Compliance, a U.S. Federal **law enacted to create and maintain standards enabling the accessibility of electronic and information technology (e.g.,, web-based content or multimedia such as portable document formats) to those with disabilities**...*"

7. p2, line 35: "2022" should not be in parentheses

Citation format has been changed.

8. p2, line 36: "2023" should not be in parentheses

Citation format has been changed.

9. p2, line 37: "if the maps change" - I suggest you clarify here that you mean that if the colourmaps (or other presentational elements) of the FPMs change. What the maps actually show (i.e. the fire perimeters) would of course be expected to change as new/updated data is included.

Great catch, and you are correct that I meant if the "colourmaps" change, as all else equal, merely changing the colourmap could lead to a change in interpretation, or misinterpretation and confusion. Revised to be explicit about colourmaps and refer back to the focus of this work on them:

"*Further, the lack of **colourmap** consistency in FPMs (cf. Figure A1 and those in Chen et al., 2022, Kochanski et al., 2023, and Liu et al., 2024) represents another potential limiting factor in user cognition for maps already displaying complex information Bunch and Lloyd, 2006) if the **colourmap** change from day-to-day, incident-to-incident, or across applications.*"

10. p2, line 42: "I" not "we"

Change made, thank you.

11. p2, line 46-47: Reference should be in parentheses

Parenthesis added, thank you for catching this.

12. p2, line 50: "I" not "we"

Change made, thank you.

13. p3, Figure caption: I suggest that the caption is expanded to include more details about what is being shown in Figure 1. However please see my general comments above for further suggestions about Figure 1.

Agreed and thank you for the suggestion. Please see the response above indicating the revised captions on Page 5.

14. p4, line 69: "fires" not "fire"

Thanks for catching this, revised to 'fires'.

15. p4, line 71-72: "The four suggested colormaps (Figure 1), one of which is an example in GeoOps (Figure A1d), intend to provide a starting point." - It is not immediately clear why Figure 1 is stated to show (only) four suggested colormaps rather than six. However please see my general comments above for further suggestions about Figure 1.

I am not entirely sure why I wrote four initially, most likely I confused myself by writing around an earlier figure design...nonetheless, this has been revised to three colormaps following the excellent suggested revisions from both reviewers 1 and 2 leading to the convergence regarding focusing on these three. New text:

"...*the **three** suggested colourmaps (Figures 1 and A2) intend to provide a starting point.*"

I look forward to seeing this published in due course!

Thank you again for your review and constructive comments!